# Cationically Polymerized Epoxy and Radiation-Cured Acrylate Blend Nanocomposites Based on WS$_2$ Nanoparticles Part B: Mechanical and Physical Properties

Gilad Gershoni [1], Hanna Dodiuk [1], Reshef Tenne [2] and Samuel Kenig [1,*]

1   Department of Polymer Materials Engineering, Shenkar College, Ramat Gan 5252626, Israel
2   Weizman Institute of Science, Rehovot 7610001, Israel
*   Correspondence: samkenig@shenkar.ac.il

**Abstract:** The radiation curing paradigm of opaque WS$_2$ nanoparticle (NP)-based epoxy/acrylate nanocomposites was studied and found to exhibit both a reduction in viscosity and an enhanced degree of curing when incorporating WS$_2$ NPs. The objective of this study was to investigate the mechanical, thermal, and physical properties of a radiation-induced and cured epoxy/acrylate blend containing 0.3 to 1.0 wt.% WS$_2$ NPs. Experimental results indicate that the tensile toughness increased by 22% upon optimizing the NP content compared to that of WS$_2$-free formulations. Tensile fractured surfaces with different WS$_2$ NP contents were analyzed with a scanning electron microscope and an atomic force microscope and showed distinctive morphology depending on the WS$_2$ NP content, supporting the results of the tensile test. The energy required to break shear adhesion specimens demonstrated an increase of up to 60% compared to that of the neat resin. The glass transition temperature determined by dynamic mechanical analysis presented similar or higher values upon WS$_2$ NP incorporation. Furthermore, up to 80% improvement in impact strength was demonstrated when WS$_2$ NPs were dispersed in the epoxy/acrylate blend. It was concluded that the surface chemistry and dispersion level of the WS$_2$ NPs are the major variables affecting the macro properties of cationically radiation-cured resins and their adhesion properties. This study is the first to demonstrate the possibility for radiation-induced curing of opaque NPs based on WS$_2$ that serve as both a reinforcement nanoparticle at low concentrations and an enhancement of the degree of curing.

**Keywords:** cationically polymerized epoxy; acrylate; nanocomposites; tungsten disulfide fullerenes; radiation-induced curing

## 1. Introduction

Epoxy resins are widely used in a variety of industries due to their inherently good mechanical properties and high adhesion Characteristics. Cationic curing (CC) of epoxies based on the starting action of photo-initiators offers advantages compared to thermal curing of epoxies, such as low energy for curing, rapid curing times, elimination of solvents, and polymerization upon demand. CC epoxy cures through a ring-opening mechanism (ROP) [1–7] that can be initiated by radical or cationic moieties [8,9] generated by photo-cleavage of photo-initiators (PI). The photocleavage products generate a cationic moiety, which initiates the ROP of the epoxy. The CC process is highly selective [8,10] as a result of the absorbance selectivity of the PI. The mechanisms of the curing by the PI were depicted in the 1970s [11,12]. Due to inherent advantages of CC many studies were dedicated to augmenting the data base of monomers and PI [1,5,7,10,13–16], leading to its use in adhesives, coatings, dental materials, and 3D printing [17–20]. To enhance their toughness, nanoparticles (NPs) were incorporated into the CC epoxy resins. Reinforcing thermally cured epoxies by NPs has been studied in the past [19–23]. Conversely, CC of epoxy nanocomposites (NCs) is a relatively recent development compared to thermally cured epoxies [2,3,19,20,24–27]. Toughening by NPs is realized by modifying the fracture

mechanisms [28] by the dispersed NPs. Toughening is analyzed by thermal and mechanical analyses as well as electron microscopy and atomic force microscopy [28–30]. Toughening of epoxy-based CCNCs has been studied via impact or tensile properties [24,25] in addition to the tribological properties [31]. It should be emphasized that the dispersion quality of the NPs affects the mechanical and thermal properties, including the toughness of the resulting NCs [29,30,32]. Dispersion of the NPs is usually accomplished by high-shear mixing, three-roll milling, sonication, and more [33]. In several studies, combinations of dispersion techniques were employed in order to achieve maximum dispersion quality. According to different reports, modification of the NPs surface chemistry may also affect the NCs final properties and long shelf life [2]. Some reports showed that improved compatibility of the NPs achieved through surface modification led to an improved dispersion quality but to inferior mechanical properties of the NCs. In several reports, in-situ preparation of the NPs during the resin-curing process led to their enhanced dispersion in the matrix [2,3]. Improvement of the mechanical properties by dispersing NPs has been well researched, but significant challenges still remain, specifically for radiation-cured NCs.

The challenges of radiation-curing NCs are complex due to the requirement that the irradiation penetrate through the entire volume of the resin film. In the case of a commercial PI this is accomplished through the tuning of the PI concentration and thickness of the photocured film. However, in the case of using NPs for the photocuring, the challenges are appreciably greater. First, the NPs should be as low-agglomerated as possible, or else the film will become optically non-uniform with abundant optical hotspots and consequently non-uniform polymerization of the film. Furthermore, the strong light absorption and scattering by the individual NPs requires that their concentration and film thickness must be carefully tuned. In fact, the selected concentration may not be the optimal one for simultaneously executing the photocuring process and achieving the reinforcement effect of the matrix. Also, keeping viscosity of the resin with the NPs in acceptable levels for molding/shaping of the final products is yet another challenge. Finally, minimizing the content of NPs in order to resolve the above-mentioned shortcomings may result in poor physical properties of the nanocomposite.

Once realizing how to overcome these challenges, significant property improvements could be achieved by either introducing pristine or surface-treated NPs. Previously published papers on tungsten disulfide ($WS_2$)-based NCs reported a significant impact improvement accompanied by a higher *Tg* [28,32,34,35]. Owing to their small size, strong light absorption, and high free carrier mobility, $WS_2$ NPs are potent oxidizing agents with respect to different photochemical reactions [36]. Since radiation-cured resins are of importance, it was perceived that the opaque $WS_2$ NPs can have a detrimental effect on their photocuring. Given their known favorable influence on the mechanical and thermal stability of different polymers, the masking effects of the $WS_2$ NPs were studied, and unexpectedly, it was found that $WS_2$ fullerenes enhanced the degree of conversion (DC) when incorporated at optimized levels in radiation-induced polymerization of epoxy/acrylate formulations (36). This finding is the first to be reported for opaque $WS_2$ in radiation-induced cured polymer systems. These results led to the present study aimed at investigating the tensile toughness, impact, and adhesion properties of $WS_2$ NCs based on epoxy/acrylate CCNCs.

## 2. Materials and Samples Preparation

### 2.1. Materials

Epoxy resin (EPV 3420TX) was supplied by Polymer G, designated as PGE. EPV 3420TX without thermal additive was also supplied by Polymer G and will be designated as PGEnTA. The resins were composed of aliphatic epoxy (45–50 wt.%), methyl acrylate (8–10 wt.%), epoxy acrylate (15–18 wt.%), polyester polyol (15–18 wt.%), and fumed silica NPs (4–6 wt.%). A special tri-photo-initiator blend (3–5 wt.%) was used, consisting of sulfonium-based cationic PI, radical PI, and thermal cationic initiator. The latter was removed for the PGEnTA resins. As can be seen, the resin is a hybrid epoxy/acrylate grade. The acrylate portion was designed for rapid initial curing and the addition of fumed silica

NPs was aimed to control the viscosity acrylate resin. These resins were designed to cure in 395 nm wavelength irradiation by LED.

Inorganic fullerenes (IF), which are composed of hollow multi-layer nanoparticles and abbreviated as IF-WS$_2$ NPs, were obtained from two suppliers, WS$_2$ NPs: WS$_2$-TO (prepared at the Weizmann Institute of Science in Israel). These NPs were synthesized and used in Part A of this study [36]. The NPs have an average diameter of 80 nm with close to a spherical shape and a hollow core. WS$_2$-C was purchased (M K Impex Corp, Canada). Sequential scanning electron microscopy (SEM) and powder X-ray diffraction (XRD) analyses revealed that these NPs had an oval shape with many irregularities (defects) and the interlayer spacing was 6.22 Å. The average particle size specified by the manufacturer is 90 nm (MKN-WS2-090).

The NPs were used as received and analyzed by a variety of methods, as summarized in Table 1.

**Table 1.** Intrinsic properties of fullerene-like nanoparticles.

| Comparison Criteria | WS$_2$-TO | WS$_2$-C |
| --- | --- | --- |
| Geometry (*d*-space-interlayer layer spacing according to XRD) | Spherical ($2\theta = 14.1°$; 6.26 Å) | Oval ($2\theta = 14.2°$; 6.22 Å) |
| Diameter | 80 nm | 90 nm |
| Moisture Content | 6.7% weight loss | 1.5% weight loss |
| pH value | 4.9 | 7.2 |
| Oxygen/Tungsten ratio by XPS | 0.55 | 0.83 |

### 2.2. Dispersion and Distribution Techniques

Two distinct procedures were utilized to study the dispersion and distribution of the WS$_2$, comprising sonification/vortex and masterbatch preparation. Results were evaluated by optical microscope (Coolpix MDC Lens by Nikon Japan). The optical microscope resolution can identify NPs of 0.8 μm in size.

### 2.3. Sonication/Vortex Multistage Dispersion Technique

The NPs were initially ground by mortar and pestle and then added to the resin in the desired quantity. A multistage combination was used employing a high-intensity horn sonicator (Q700, Qsonica L.L.C, Newtown, CT, USA). Sonication resulted in good dispersion. The distribution was accomplished by intensive vortex mixing (Wizard IR Infrared Vortex Mixer, VELP Scientifica, Usmate, Italy) for a duration of 2 min at 3000 RPM. Ice-cooling was employed during the multi-stage dispersion procedure in order to keep the viscosity as high as possible under the high shearing rates used for the dispersion of the NPs. Ice-cooling was also exercised for preventing pre-curing of the resins during dispersion. Several repetitive stages were applied to enhance the dispersion quality.

Dispersion and distribution techniques were described in an earlier report [36].

### 2.4. Curing System

Curing was conducted with 395-nm wavelength LED irradiation. Curing of all the samples, excluding the shear specimens, was conducted in a transparent silicone mold (SORTA-Clear 40, Smooth-On, PA, USA) with a 2-mm-thick transparent silicone cover. Schematic illustration of this system is given elsewhere [36]. The distance of the mold from the LED was ~40 mm. The LED radiation intensity was up to 7 W/cm$^2$.

### 2.5. DMA Characterization

The as-received resins contained 4–6% fumed silica. The weight fractions of WS$_2$ NPs incorporated in the resin were: 0, 0.3, 0.5, 0.75, and 1.0 wt.%. The samples for the dynamic mechanical analysis (DMA) were $25 \times 6 \times 0.3$–0.4 mm in size and were cured for 240 s. The parameters of the DMA procedure were as follows: 1 Hz, 5 μm amplitude, and temperature

ramp of 3 °C/min in a temperature range of 0 °C to 140 °C. (DMA Q800, TA Instruments, New Castle, DE, USA).

### 2.6. Tensile Tests

Special cavities were manufactured for the preparation of dog-bone samples having a thickness of 0.3–0.4 mm. The content of the NPs in the nanocomposites was the same as those prepared for the DMA analysis. The curing cycle was 300 s. A total of 5–8 samples from each resin type and NP concentration were prepared and characterized using a universal testing machine (Instron 4481, Grove City, PA, USA) at a loading rate of 1 mm/min. Only PGE with $WS_2$-TO was tested.

### 2.7. Impact Test

Samples with a thickness of 0.5 mm were prepared using the same procedure as described above and having the same NP content. Samples were tested according to modified IZOD ASTM D-256 with 0.5 J pendulum (Resil 5,5, Ceast, Turin, Italy). Two impact samples were measured for each formulation of the PGE and 4–5 samples for each of the PGEnTA formulations.

## 3. Characterization

### 3.1. DMA Characterization

The as-received resins contained 4–6% fumed silica. The weight fractions of $WS_2$ NPs incorporated in the resin were: 0, 0.3, 0.5, 0.75, and 1.0 wt.%. The samples for the dynamic mechanical analysis (DMA) were $25 \times 6 \times 0.3$–0.4 mm in size and were cured for 240 s. The parameters of the DMA procedure were as follows: 1 Hz, 5-μm-amplitude, and temperature ramp of 3 °C/min in a temperature range of 0 °C to 140 °C. (DMA Q800, TA Instruments, New Castle, DE, USA).

### 3.2. Tensile Tests

Special cavities were manufactured for the preparation of dog-bone samples having a thickness of 0.3–0.4 mm. The content of the NPs in the nanocomposites was the same as those prepared for the DMA analysis. The curing cycle was 300 s. A total of 5–8 samples from each resin type and NP concentration were prepared and characterized using a universal testing machine (Instron 4481, Grove City, PA, USA) at a loading rate of 1 mm/min. Only PGE with $WS_2$-TO was tested.

### 3.3. Impact Test

Samples with a thickness of 0.5 mm were prepared using the same procedure as described above and having the same NP content. Samples were tested according to modified IZOD ASTM D-256 with 0.5 J pendulum (Resil 5,5, Ceast, Turin, Italy). Five impact samples were measured for each formulation of the PGE and 4–5 samples for each of the PGEnTA formulations.

### 3.4. Single-Lap Shear

A 0.1–0.2 mm thickness of nanocomposite resin layer was applied and cured between two pre-cleaned (by EtOH and acetone) glass fiber-reinforced (GFR) epoxy plates (FR4). The FR4 thickness was 2 mm. This GFR material does not absorb radiation at 395 nm: hence, curing of the studied resins at this wavelength is applicable. The overlap length was 12.6–13 mm. The GFR plate width was 25 mm. Samples were placed ~30 mm from the LED source and cured for 18 min. The long curing cycle time was dictated by the absorbance of the relatively thick GFR plates. Measurement of the samples was conducted ~30 min after curing. Loading of the specimens was conducted according to ASTM D1002 using a loading rate of 5 mm/min. The test was employed on a universal testing machine (Instron 4481, Grove City, PA, USA).

### 3.5. SEM Analysis

The tensile test surfaces were coated with gold (SC7620, Quorum Technologies Ltd., Lewes, United Kingdom) to prevent electron charging of the specimens. The samples were analyzed by SEM (scanning electron microscope) (JSM-IT200, Jeol, Akishima, Japan). Energy-dispersive X-ray analysis (EDX) was performed, as well. The accelerating voltage ranged from 10 to 20 kV. To increase the accuracy of the EDX results, the voltage was raised from 10 to 20 KV.

### 3.6. AFM Analysis

The fractured surface of the tensile specimens was characterized with an atomic force microscope (AFM) (Bruker-Innova AFM with RESP 20 probe) in contact mode.

### 3.7. Spectrophotometry

The absorbance spectrum was measured with a UV-visible spectrophotometer from 440 nm to 360 nm (UV-1650PC, Shimadzu, Kyoto, Japan).

## 4. Results and Discussions

### 4.1. DMA

The DMA was used to determine the glass transition temperature (*Tg*). The *Tg* was evaluated using the loss-modulus or tan-delta representations. It was observed that a single *Tg* was obtained, though the resin and was composed of epoxy and acrylates. High DC of the epoxy constituent leads to higher crosslinked density, which in turn, results in higher *Tg*. It should be emphasized that the presence of the NPs may also affect the *Tg* due to possible interactions between the molecular network and the NPs. Figure 1 describes the *Tg* variations for the different resins, WS$_2$ sources, and concentrations. As can be seen in Figure 1, the *Tg*s determined according to tan-delta measurements exhibited the same tendency as the *Tg*s obtained from the loss modulus analysis.

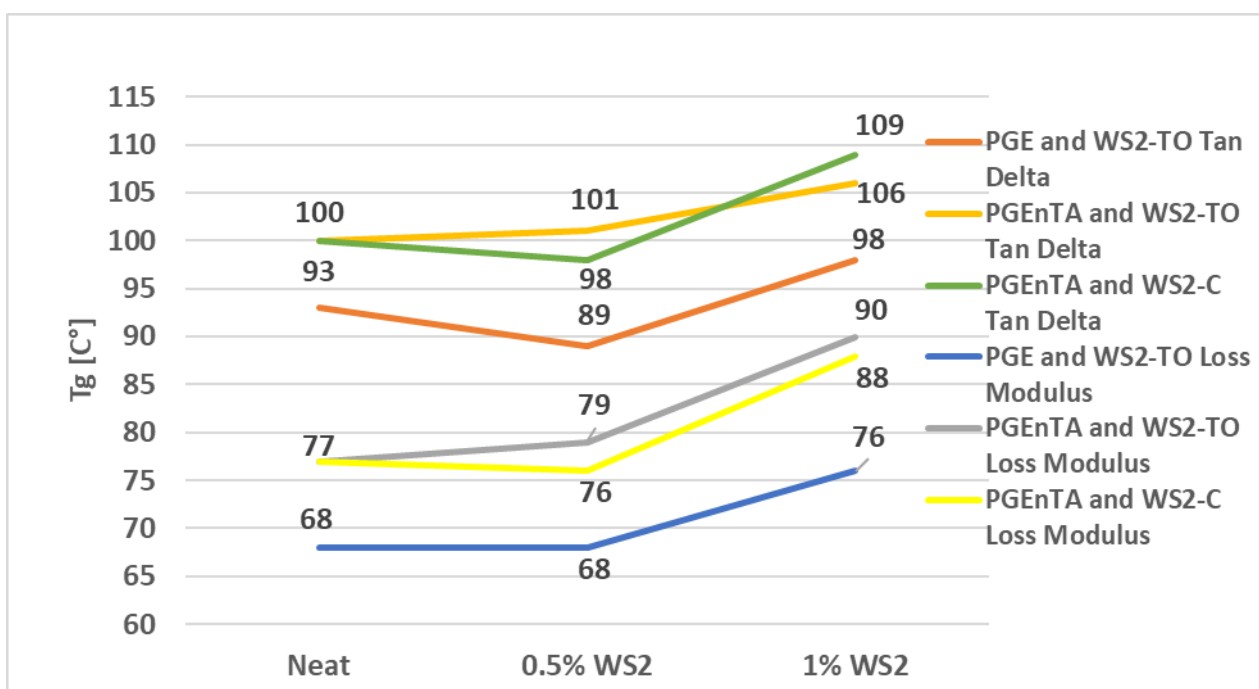

**Figure 1.** DMA analysis of neat resin and resins with NP content of 0.5 wt.% and 1 wt.%.

As can be observed from these graphs, the PGE-based resin had a lower *Tg* than the PGEnTA-based resin. This may be attributed to the thermal activation of the thermal initiator by the heating effect of the radiation. As a result, vitrification of the PGE took place

sooner, limiting the maximum crosslinking density achievable in such a curing process. Upon increasing the $WS_2$ concentration to 1 wt.%, in PGEnTA, both $WS_2$ sources showed an increase in *Tg*. The most significant increase (by 11 °C) in *Tg* was noticed for $WS_2$-C.

Based on the DMA results, a higher toughness may be expected for PGE with 0.5 and 1 wt.% $WS_2$-TO. For PGEnTA NCs, higher toughness may be anticipated for 0.5 wt.% of $WS_2$-C NCs but at 1 wt.% loading a higher toughness is envisaged for $WS_2$-TO.

### 4.2. Tensile Properties

Tensile properties were measured only for PGE with 0.3 to 1.0 wt.% $WS_2$-TO. The toughness was determined from the area under the stress–strain curves. As can be seen in Figure 2, an increase of 22% in toughness was accomplished by incorporation of 0.5 wt.% of $WS_2$. Above this, optimal concentration the toughness decreased. This may be attributed to agglomeration, which led to even lower toughness in the case of 1.0 wt. % of $WS_2$, compared to the neat resin.

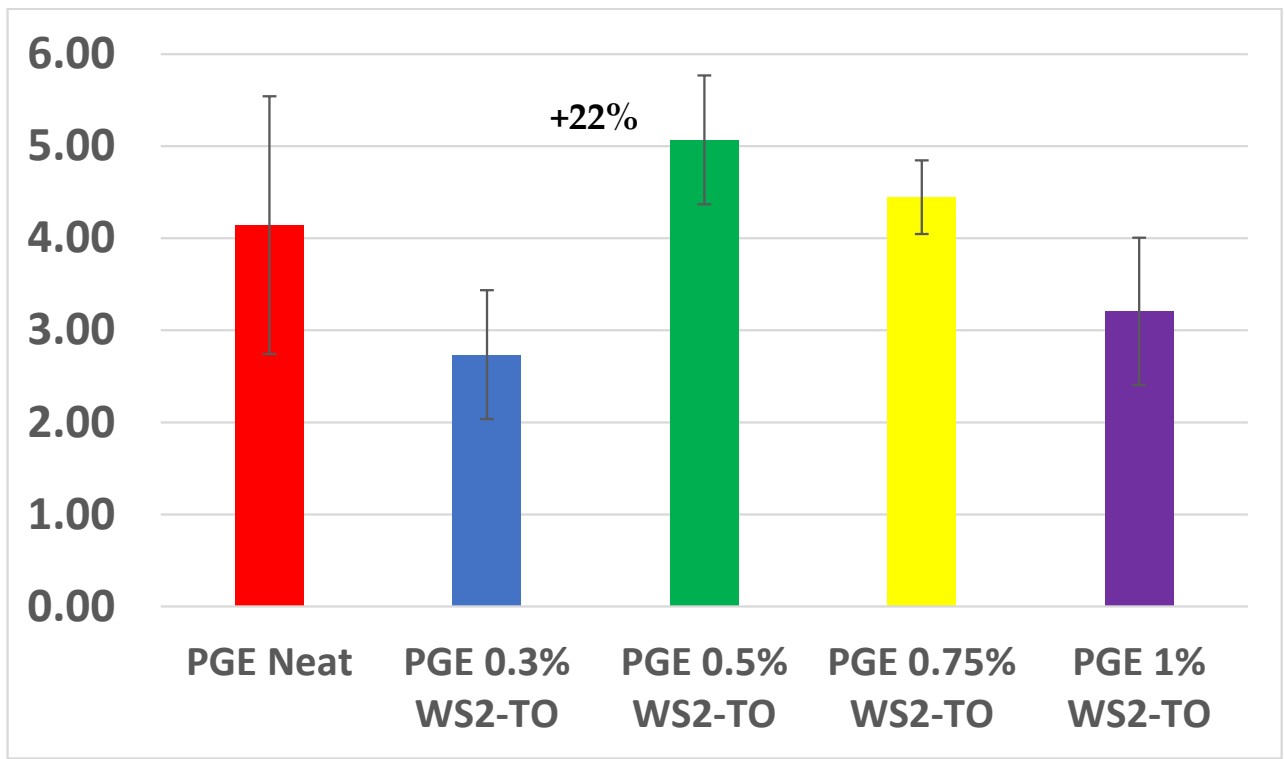

**Figure 2.** Results of tensile toughness test of PGE with various contents of $WS_2$-TO.

### 4.3. Adhesion Strength

The adhesion strength of the various compositions was investigated using lap-shear-type specimens based on glass-reinforced polymer (GRP F-4). Experimental results have shown that all the specimens failed in the FRP adherent interface, as can be seen in Figure 3 (for PGE with 0.5 wt.% $WS_2$-TO).

Hence, it was concluded that since the resin showed stronger adhesion to the GRP substrate, the adhesion strength of the NCs adhesives could not be determined accurately with these substrates. Nonetheless, strain-to-failure and energy-to-break were determined for the formulations studied. Results of the single-lap shear specimens can be observed in Figures 4 and 5.

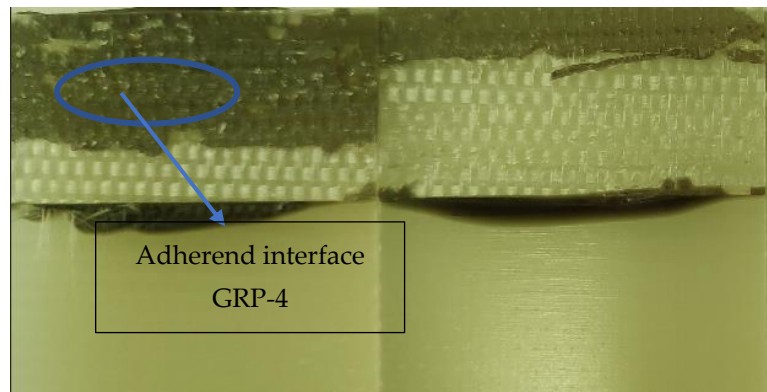

**Figure 3.** Optical microscope image of fractured surfaces of lap shear specimen of PGE with 0.5 wt.% WS$_2$-TO.

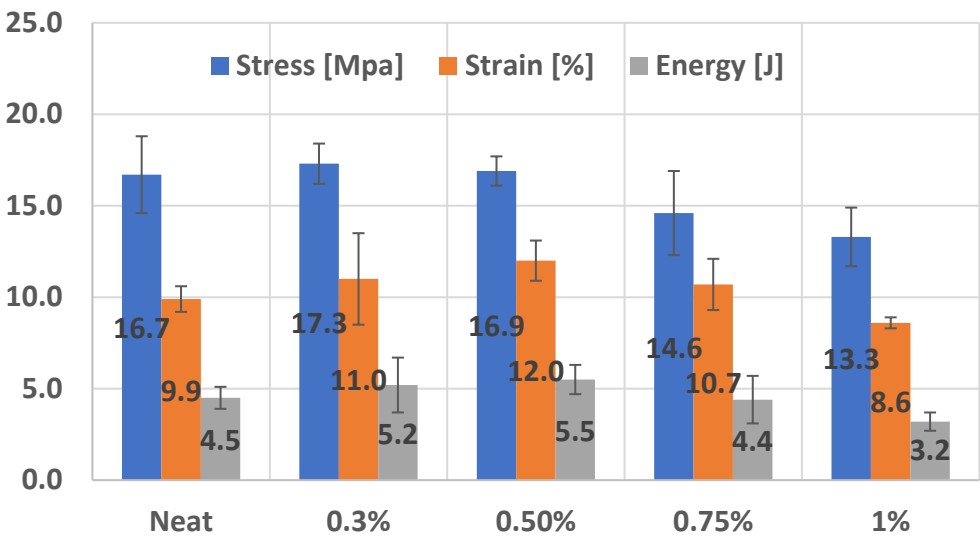

**Figure 4.** Single-lap shear of PGE with various contents of WS$_2$-TO.

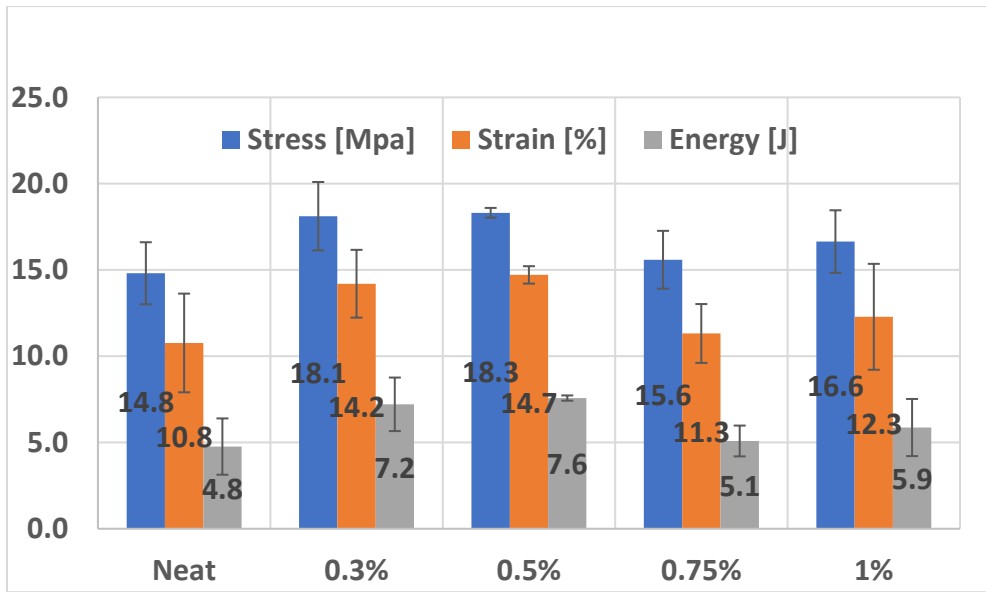

**Figure 5.** Single-lap shear of PGEnTA with various contents of WS$_2$-TO.

As evident from the results, PGEnTA attributes were higher than those of PGE samples for all formulations, excluding the strength values for the neat samples. A possible explanation for this observation is the slower curing kinetics of PGEnTA compared to the fast curing of PGE, which may lead to higher crosslink densities and greater uniformity of the resin film. PGEnTA-based NC adhesives exhibited a higher level of failure for WS$_2$-C-containing formulations compared to WS$_2$-TO ones. PGE-based compositions showed 100% cohesive failure in the adherents. Results indicate that there was no clear difference between 0.3, 0.5, and 0.75 wt.%-containing WS$_2$-C resins; all three compositions displayed superior properties compared to neat and 1% NP-containing samples. Higher variability in the results were present in NCs based on PGEnTA with WS$_2$-C compared to PGEnTA with WS$_2$-TO samples. This may be the result of mixed cohesive and adhesive failure mechanisms. Finally, PGEnTA containing WS$_2$-TO at 0.3 wt.% and 0.5 wt.% concentrations showed a clear superiority in adhesion strength.

The energy at break demonstrated a 20% increase for PGE-based compositions (Figure 5). In the case of PGEnTA, a 40% increase for WS$_2$-C and 60% increase for WS$_2$-TO NPs were obtained (see Figures 5 and 6). As can be observed, the energy-to-break decreased at 1.0 wt.% of the WS$_2$.

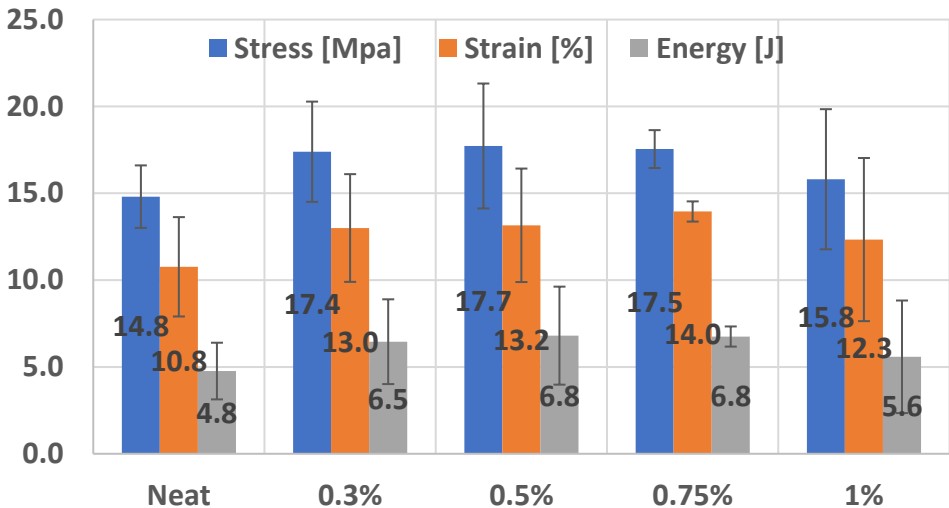

**Figure 6.** Single-lap shear of PGEnTA with various contents of WS$_2$-C.

*4.4. Impact Properties*

As can be seen in Figure 7, PGEnTA exhibited 25% lower impact resistance than PGE. This may be attributed to the thermal additive (TA) effect on the crosslinking density of the various compositions, which led to increased strength of PGEnTA over PGE, resulting also in reduced toughness.

Distinctively, as the WS$_2$-TO content was increased above 0.5 wt.% a significant decrease in the impact strength of the PGE formulations could be perceived, in contrast to a moderate decrease in the case of PGEnTA-based NCs. PGEnTA-based NCs exhibited higher impact strength for higher NP content, in addition to higher *Tg*, compared to PGE based-NCs. In both resin systems, the impact strength decreased as the concentration of the WS$_2$ NPs approached 1.0 wt. %, as was the case for the toughness and energy-to-break (Figures 2 and 4–6).

As can be seen in Figures 7 and 8, the best impact results were obtained for 0.5% content for both WS$_2$ sources. Significant impact improvement of 75% for PGEnTA and 60% for PGE were achieved at optimal concentration of the WS$_2$ NPs.

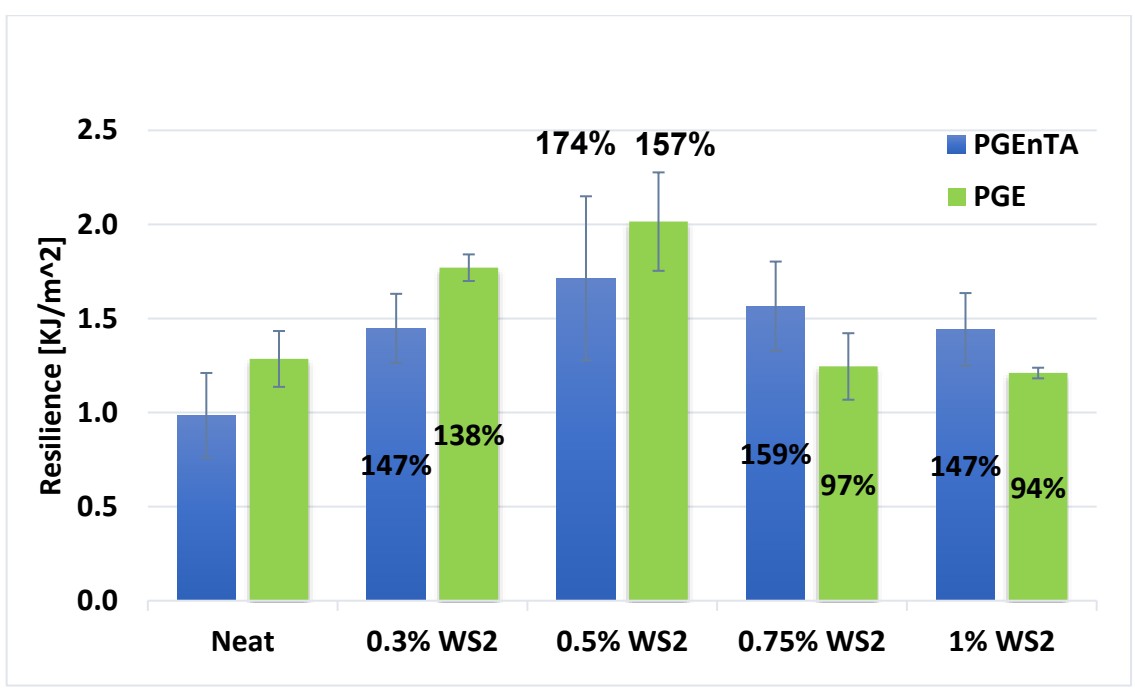

**Figure 7.** Impact strength of PGE and PGEnTA with WS$_2$-TO.

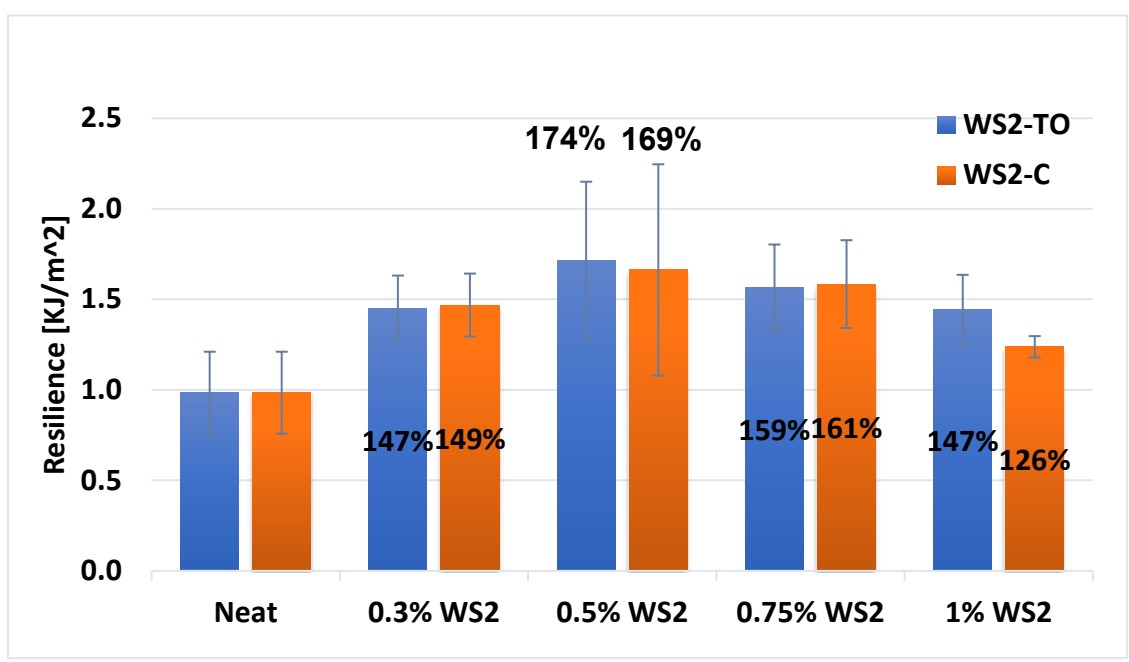

**Figure 8.** Impact strength of PGEnTA with WS$_2$-TO and WS$_2$-C.

*4.5. Effect of NP Content on the Mechanical Properties*

From the results obtained in tensile, impact, and shear tests, it can be concluded that overall, the optimal results and improvements of 22%, 75%, and 60%, respectively, were reached with 0.5 wt.% of WS$_2$-TO. Thus, one can conclude that the addition of WS$_2$-TO (IF-WS$_2$ NPs) to the resin has beneficial effect, not only on the curing kinetics [36], but also on the mechanical properties of the NCs. Adding WS$_2$-C led to improvements of 69% and 40% in the impact and shear, respectively. Surface chemistry analysis suggests that the higher oxygen content of the WS$_2$-C inhibits the positive effect of NPs reached by the WS$_2$-TO and could be the reason for the overall superiority of the WS$_2$-TO in improving the behavior of the photocured epoxy resin. The WS$_2$-TO exhibits larger interfacial interaction

between the NPs and the resin, since its higher sulfur content leads, at the surface (smaller average size), to more readily bonding with the resin matrix, as discussed previously [36].

The mechanical properties of the NCs are compromised upon increasing the content of the NPs beyond this threshold, even below the values of the neat sample, as was evident for impact results of 0.75 wt.% and 1.0 wt.% of WS$_2$- TO in PGE (Figure 7). This shows the complex nature of higher loadings of the WS$_2$-IF, which favorably affects the curing kinetics [36], but impairs the mechanical properties. Indeed, agglomeration of the NPs is often observed beyond 0.5 wt.% loading of the IF NPs.

### 4.6. Analysis of the NC Surfaces Fractured by Impact and Tensile

SEM analysis was carried out in order to study the fracture mechanisms and the mechanical properties of the radiation-cured nanocomposites (RCNCs) based on cationic curing of the epoxy and radiation curing of the acrylate using the tri-initiator system. Figure 9 shows the basic difference between neat and RCNCs systems, based on PGE.

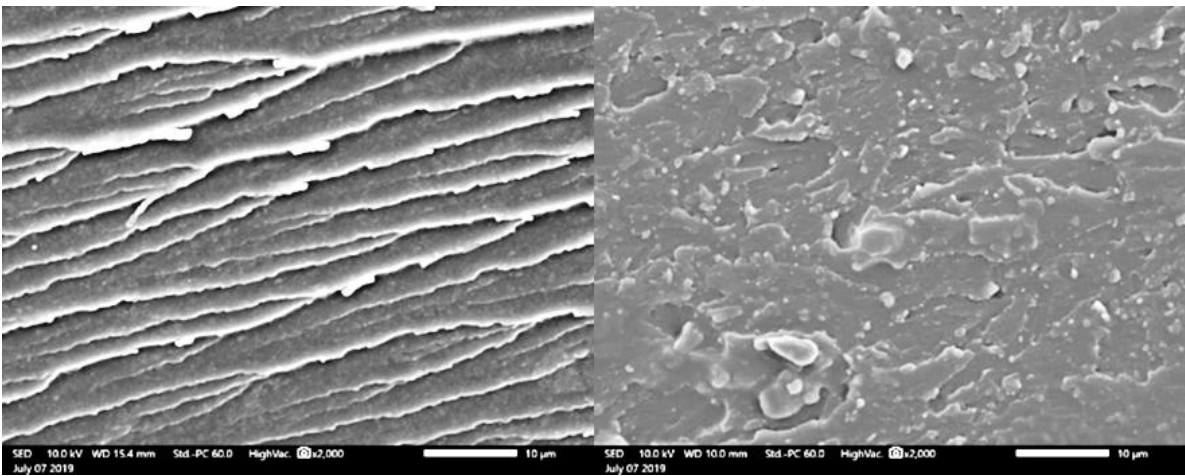

**Figure 9.** SEM micrographs of neat PGE (**left**) and PGE with 0.3% WS$_2$-TO (**right**).

While the neat system displays a brittle fracture, the NC surface demonstrates roughness and a ductile type of fracture. These observations support the fundamental mechanism that affects the enhancement of the mechanical properties and especially the energy absorption-related properties, such as impact energy-at-break, for both shear and tensile loading.

Figure 10 depicts the detailed morphology of the NP-reinforced specimens following tensile testing.

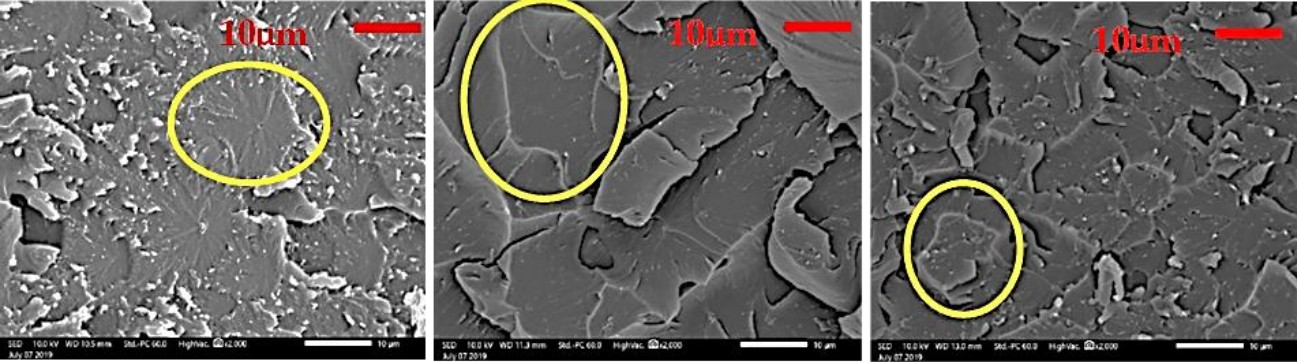

**Figure 10.** SEM micrographs of tensile-fractured surface of neat PGE (**left**), 0.5 wt.% of WS$_2$-TO in PGE (**center**), and 1 wt.% of WS$_2$-TO in PGE (**right**). Circled areas show a typical crater morphology found in the fractured surface.

As can be noticed in Figure 10, craters with nodular morphology were formed during loading-to-failure in all the samples. However, the craters differed in size and boundary lines. It should be emphasized that in the middle of each crater a nanoparticle or an agglomerate of such NPs is situated. This is also the case for compositions that contain only silica NPs. Analysis of the results indicate that enhanced mechanical properties were obtained when larger craters and sharper outlined borderlines were observed. The $WS_2$ NP- containing formulations (in addition to the nano-silica) revealed larger craters and hence enhanced properties. It may be postulated that the nodular boundaries could be induced during failure by crack deflection, which nucleated at the NPs. The larger craters were obtained at $WS_2$ concentration of 0.5 wt.%. Increasing the concentration to 1.0 wt.% led to reduced crater size and reduced properties. The size of the crater is the result of the energy expended in forming the crater. Hence, larger craters indicate higher level of energy dissipation. EDX analysis was carried out in order to evaluate the elemental composition of the NPs at the centers of the nodules. $WS_2$ was found in the center of the nodules in 0.5 wt.% and 1.0 wt.% $WS_2$, as can be observed in Figure 11. In the case of neat resins, craters were also noticed.

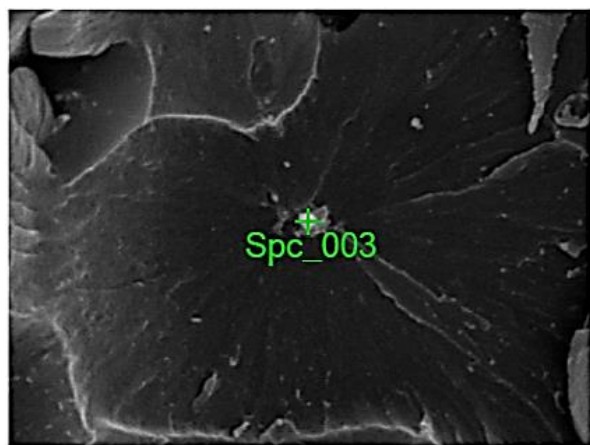

| Display name | Standard data | Quantification method | Result Type |
|---|---|---|---|
| Spc_003 | Standardless | ZAF | Metal |

| Element | Line | Mass% | Atom% |
|---|---|---|---|
| C | K | 30.58±0.29 | 75.17±0.71 |
| O | K | 4.06±0.17 | 7.50±0.31 |
| Si | K | 3.98±0.10 | 4.18±0.11 |
| S | K | 2.03±0.10 | 1.87±0.09 |
| Cl | K | 1.45±0.07 | 1.21±0.06 |
| Pd | L | 9.18±0.25 | 2.55±0.07 |
| Sb | L | 1.18±0.14 | 0.29±0.03 |
| W | M | 10.81±0.37 | 1.74±0.06 |
| Au | M | 36.72±0.48 | 5.50±0.07 |
| Total | | 100.00 | 100.00 |
| Spc_003 | | | Fitting ratio 0.0821 |

**Figure 11.** EDX result of 0.5 wt.% $WS_2$-TO in PGE.

An AFM study was carried out to supplement the SEM investigations. The AFM analysis was conducted in contact mode, scanning an area of $50 \times 50$ μm.

As evident from Figure 12, the fractured surface of the resin containing silica NPs exhibited small sporadic nodules alongside needle-like asperities. The 0.5 wt.% $WS_2$-containing NCs exhibited a rough surface, with a high density of large and small craters having clear borders. The 1.0 wt.% $WS_2$ NC morphology was a combination of the neat and 0.5 wt.% $WS_2$ samples.

Table 2 summarizes the measured roughness of the fractured surfaces shown in Figure 12. Here, Rq is the root mean square roughness, Ra is the average roughness, and Rmax is the maximum roughness depth, all within the area of 50 μm $\times$ 50 μm measured.

As evident from Table 2, the higher the statistical roughness parameters, the higher were the tensile toughness, impact strength, and energy-to-failure in shear. The mechanical properties as well as the statistical roughness parameters attained their maximum at 0.5 wt.% of $WS_2$.

The specific crater dimensions, i.e., length, depth, and border height, were further analyzed by AFM and showed significant differences between the samples, as can be seen in Figure 13 for the 0.5 wt.% of $WS_2$-containing sample.

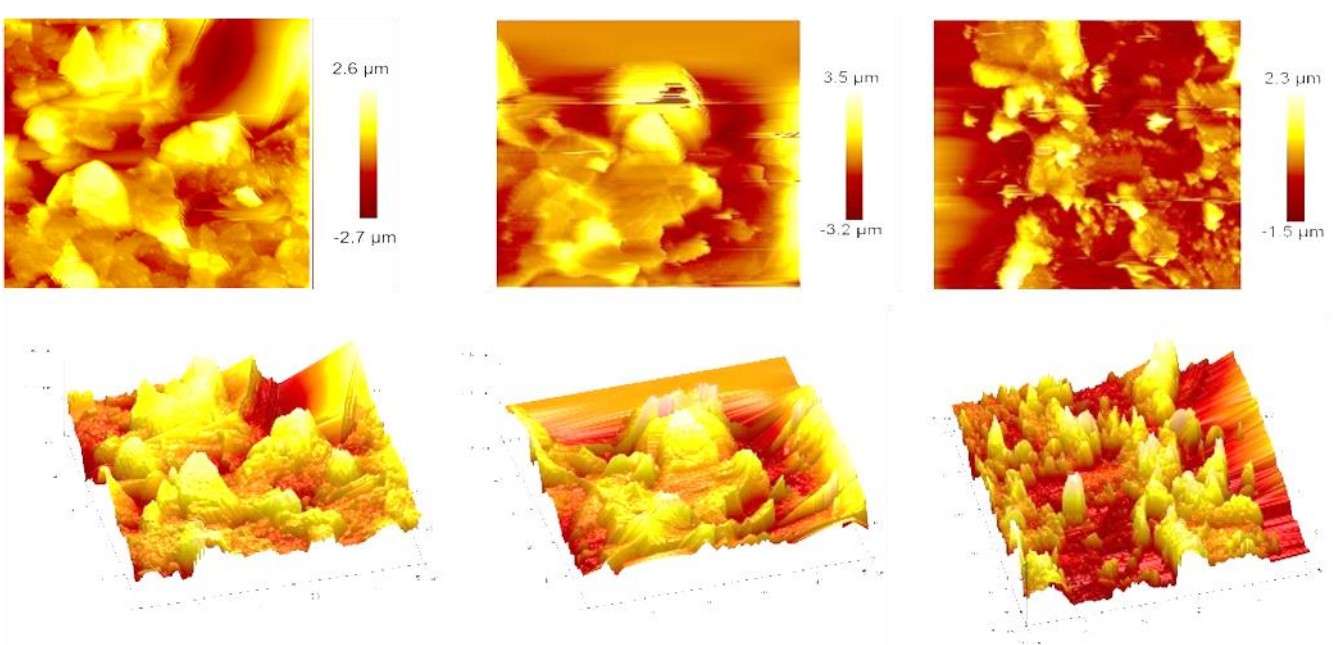

**Figure 12.** Contact mode AFM scan of 50 μm × 50 μm area of the surface of tensile fractured neat PGE (**right**), 0.5 wt.% WS$_2$_TO in PGE (**middle**), and 1 wt.% WS$_2$-TO in PGE.

**Table 2.** Surface roughness of a neat PGE (without WS$_2$), PGE with 0.5 wt.% WS$_2$-TO, and PGE with 1 wt.% WS$_2$-TO.

| Sample | Rq | Ra | Rmax |
|---|---|---|---|
| PGE with silica NPs | 0.55 μm | 0.43 μm | 3.87 μm |
| PGE with 0.5 wt.% WS$_2$-TO | 0.90 μm | 0.65 μm | 7.93 μm |
| PGE with 1 wt.% WS$_2$-TO | 0.76 μm | 0.60 μm | 5.28 μm |

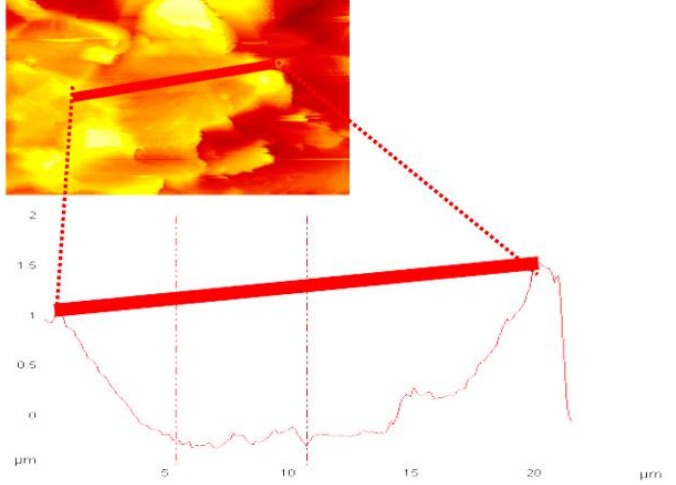

**Figure 13.** AFM nodule size morphology analysis of PGE with 0.5 wt.% WS$_2$-TO, as received by the AFM system.

The same analysis was done for neat PGE and PGE with 1.0 wt.% WS$_2$-TO. The results are summarized in Table 3.

**Table 3.** Nodule morphology of PGE without $WS_2$, PGE with 0.5 wt.% $WS_2$-TO, and PGE with 1 wt.% $WS_2$-TO.

| Sample | Nodule Length | Nodule Depth | Nodule Border Height |
|---|---|---|---|
| PGE with only silica NPs | 430–630 nm | 6.5–11.5 µm | 200–350 nm |
| PGE with 0.5 wt.% $WS_2$-TO | 1350–1800 nm | 20 µm | 1100–1300 nm |
| PGE with 1 wt.% $WS_2$-TO | 900 nm | 9 µm | Not Defined |

Since the craters are formed during the fracture process as a result of crack deflection, the larger the crater the more energy is being dissipated during fracture, leading to enhancements in energy-related properties such as impact, tensile, and shear energy-to-break. As can be distinguished in Table 3, the mechanical properties show an optimal value at 0.5 wt.% of $WS_2$ NPs, beyond which they are adversely affected by agglomeration and craters' overlap. As stated above, nodules can also be generated by the silica NPs present in the neat resin.

Compared to a previously published paper on the fracture mechanism-induced morphology [31] of $WS_2$ NPs, no cavitation was observed. This may lead to the assumption that proper surface treatment of the $WS_2$ may result in even higher toughness results, provided cavitation could be achieved.

## 5. Conclusions

IF-$WS_2$ NPs have a significant potential for nanocomposites based on cationic polymerized epoxies/radiation-cured acrylates. Careful deliberation on the effect of the NPs on the photocuring process and the overall physical properties of the nanocomposite is provided in the introduction. Taking account of these mutual effects simultaneously has been shown to lead to an enhanced photocuring process of the resin matrix (part A) simultaneously with large improvements in the mechanical properties of the resin film studied herein. The main effect of the $WS_2$ NPs is the substantial increase in the energy absorption during impact loading, which leads to 80% and 60% increase in the shear adhesion strength. SEM and AFM of fractured surfaces indicate that distinctive morphology was developed depending on the level of loading with the $WS_2$ NPs, supporting the mechanical test results. The glass transition temperatures ($Tgs$) were similar or higher upon $WS_2$ NP incorporation. It was found that the surface chemistry and dispersion techniques of the $WS_2$ NPs are the major variables affecting the bulk properties of cationically cured resins and their adhesion properties. Furthermore, the failure mechanism is affected by the compatibility of the $WS_2$ NPs and the resin. It could be concluded that by better understanding the effect of the $WS_2$ NPs on the photo/cationically cured systems, better and more tunable system design could be achieved for the unique radiation-cured epoxy/acrylate containing $WS_2$ NPs. This study is the first to demonstrate the possibility for radiation-induced curing of opaque NPs based on $WS_2$ (Part A) that serve as both a reinforcement nanoparticle at low concentrations and an enhancement of the degree of curing.

**Author Contributions:** Conceptualization: H.D., S.K., R.T.; Investigation: G.G.; Writing: G.G. Review: S.K., R.T.; Supervision: H.D., S.K., R.T. All authors have read and agreed to the published version of the manuscript.

**Funding:** Innovation Authority of Israel and the Fraunhofer (Germany) research program, grant no. 64772.

**Acknowledgments:** RT would like to acknowledge the following foundations: The estate of Manfred Hecht and the estate of Diane Recanati. He is also grateful to the Irving and Cherna Moskowitz Center for Nano and Bio-Nano Imaging, the Perlman Family Foundation, and the Kimmel Center for Nanoscale Science.

**Conflicts of Interest:** The authors declare no conflict of interest.

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
