# Peer review of "Cationically Polymerized Epoxy and Radiation-Cured Acrylate Blend Nanocomposites Based on WS2 Nanoparticles Part B: Mechanical and Physical Properties"

_jcs, doi:10.3390/jcs7010042_

Round 1
Reviewer 1 Report
The authors reported the investigation of the cationic polymerization for the epoxy-WS2 and silica particle composites. Epoxy resins are necessary materials that should be improved in terms of the formation of composites and their methodologies. However, this work has shown the standard photo-curing methods with pristine WS2 pillars or silica particles. The scientific conclusion seems narrowly focused on the specific combination of the particles that could be potentially generalizable. Therefore, I am not supportive of suggesting the publication of this manuscript in J. Compos. Sci.
1. The title is not in a proper style, such as “xxx Part B: xxx”. As a result, the reviewer cannot find any relationship to other associated articles.
2. There is many error codes “Error! Reference source not found..”
3. Describe the detailed curing mechanism and relevant chemical schemes for potential readers.
4. Fig3 seems to need to highlight detailed descriptions of which points should be the significant fractures.
5. The error bars in Fig 7 and 8 seem statistically out of range, which can affect the conclusions.
6. The red line in Fig 13 should be re-considered because it is not parallel to the x-axis.
Author Response
Attached find the response

Reviewer 2 Report
As dispersion is always a challenge in NCs, some indication on the methods used to disperse the added NPs would be required. It is mentioned that the methods are reported in a previous work (Ref 108), but this reference it is indicated as to be published.
In line 200, "This may be the result of mixed failure mechanism". If there is any evidence of which mechanisms, it should be commented.
Finally some minor writing comments:
-Some abreviations are not defined, such as IF in line 89 or RCNCs in 240 & 242.
-It is not recommended to start a phrase with a number, such as in line 112 and 118. You coud rephrase to put the number in the middle, or put the number as a word.
Author Response
Attached please find the response

Reviewer 3 Report
This work deals with the preparation and physico-chemical characterization of radiation induced and cured epoxy/acrylate blend containing WS2 and silica NPs. The viscoelastic properties of the materials were investigated together to their mechanical properties and morphological features as a function of their composition.
I find this paper falling within the scope of the journal. However, it needs to be greatly improved because, as it is for now, the quality is low.
1.The origin of all the starting reagents should be reported.
2.the methods for the experimental preparation of the nanoparticles embedded into the organic matrix based nanocomposites should be reported in the experimental section.
3.For what concerns the tensile properties, the authors report that “The toughness was determined from the area under the stress-strain curves.” (line 167). This properties is not clear, it should be referred to it as “stored energy”, as a result of curves integration. Please check.
4.It is recommended to add to the MS also the stress vs. strain curves and discuss any trend concerning the Young’s Modulus (elastic modulus), as one of the most crucial parameters to study tensile properties of nanocomposites.
5.There are different morphological errors in the writing. I suggest to let the MS be read by an english native speaker.
6.Please improve the conclusions, by adding more experimental findings.
7.“Error! Reference source not found..” is present many times within the text (e.g. lines 92,151,176,268, 300, etc..). It gives the idea that the paper has never be read before submission. Also, the style format changes many times.
8.The introduction should be updated with the most recents paper about the use of inorganic nanofillers for the preparation of nanocomposites, and their effect on the thermo -, viscoelastic and mechanical properties (please see: https://doi.org/10.3390/jcs6070182, https://doi.org/10.3390/jfb9040060 and https://doi.org/10.3390/jcs6110333).
Author Response
Attached please find the response

Reviewer 4 Report
The article by Gershoni G. et al. studies the mechanical properties of epoxy/acrylate systems containing silica and tungsten sulfide particles. The authors evaluate how the addition of WS2 particles affects the glass transition temperature and tensile/impact strength of formulations already containing silica, and then they associate the change in properties with a change in the morphology of the fracture surfaces. On the whole, the article is very crude and careless, with many typos, misspellings, and slang expressions. The essence of the work is little clear. The work seems chaotic, as some of the research is done on only one series of samples and the other two series are left out and there is no clear comparison of the series and their role and purpose. The authors use complex multicomponent systems of unknown composition without a proper explanation of the reasons for their complexity. There is no scientific value in the article, and the practical value is not clear. The English grammar in the article is also bad. On this basis, I cannot recommend the article in its current form for publication in JCS.
Some other specific comments are as follows.
Title: “…nanocomposites based on WS2 and silica nanoparticles”. The authors write about nanoparticles but provide no evidence. Particle size and its distribution must be determined for both silica and tungsten sulfide, e.g. by using DLS.
Lines 37, 38, 40, 41: “[13]–[41]”, “[42]–[55]”, “[56]–[78]”, “[79]–[98]”. The authors cite 82 (!) references without any detailed review of these works. It is not acceptable. The authors should reduce the number of references here to a maximum of 7-10, giving only references to key reviews. At the same time, the Introduction itself is very short and does not reflect the theme of the article. For example, it does not reflect the following questions. What is the reason for using WS2? Why is it in the form of fullerenes or platelets? What are the peculiarities of both? Why is WS2 used in combination with silica? Why is an epoxy/acrylate mixture used? The introduction should be expanded according to the necessity of answering these questions.
Lines 77-80: “aliphatic epoxy (45-50 wt.%), methyl acrylate (8-10 wt.%), epoxy acrylate (15-18 wt.%), polyester polyol”, “sulfonium-based cationic PI, radical PI and thermal cationic initiator”. The authors should provide information on the specific chemical composition of the used compounds, their molecular weight, the content of epoxy groups, and so on.
Line 89: “not IF nanoparticles” What does IF mean?
Lines 92, 151, 176, 184, 268, 282, 295, 301: “Error! Reference source not found”. There are no references here.
Line 93: “fullerene-like nanoparticles” On line 88, the authors write that WS2-C are platelet NPs, not fullerenes.
Table 1: “…d-spacing … Spherical (2θ=14.1) 6.3 Å”. How do spherical particles have an interlayer distance? What is the meaning of this value?
Table 1: “Diameter … 90 nm”. What is meant by the "diameter" of the platelets? How was it determined?
Table 1: “Oxygen/Tungsten ratio”. Why is there oxygen in WS2? If it is a surface-modified WS2, it should be explicitly written about it with an indication of the chemical structure of the grafted groups and their estimated content.
Lines 101-119: This part should be moved to section "3. Characterization".
Line 102: “The as received resins contained 4-6% fumed silica.” Why did the authors not use silica-free resin?
Line 118: “2 impact samples were measured for each formulation”. Two specimens are too few to assess mechanical properties. At least 5 specimens must be tested.
Figure 11: Why is the atomic concentration of tungsten comparable to the atomic concentration of sulfur, which should be twice as high? How did palladium, chlorine, and antimony get in the system? It looks like there is an incorrect calculation of the elemental content due to improper settings of the instrument.
Table 2. In the table, the standard deviation of the measured values should be shown. How many 50μm*50μm areas did the authors use to estimate the values reported?
Line 316: “IF-WS2 NPs have a significant potential for CCNCs based on cationic polymerized epoxies/ radiation cured acrylates”. The importance of cationic polymerization, radiation curing and epoxy/acrylate blending in relationship to the applied nanoparticles is not disclosed in the article. Also, the conclusion contains nothing about the type of WS2 particles and is essentially pointless.
Author Response
Attached please find the response. Thanks for your review

Round 2
Reviewer 1 Report
I have no comments. I don't think the authors revised the manuscript properly.
Author Response
We revised the manuscript. We would appreciate if Reviewer 1 can elaborate
Reviewer 2 Report
I do not see fully reflected my first comment in previous review about indicating dispersion method used.
A relevant part of the introduction is dedicated to describe the dispersion challenge, including several references and enumerating typically used methods, but afterwards there is not any mention to the method used in this work. The included summary of Part A indicated by the authors only refers to the finding related to the enhanced degree of conversion with the WS2 fullerenes.
As in the conclusions it is stated again that surface chemistry and dispersion techniques are key variables affecting the obtained properties, it should be included in this paper at least a short indication of the methods used and distribution level obtained, keeping the reference to Part A paper for further details.
The proper location would be completing the following sentence "Dispersion and distribution techniques were described in an earlier report [36]."
Comments on author's answers:
In line 112 the number (300) is in the middle of the sentence
5-8 samples from each resin type and NPs...
In line 118 we cannot locate the number in the sentence
5 impact samples were measured for each formulation...
Author Response
Attached please find the response to Reviewer 2

Reviewer 3 Report
The proposed corrections haven't be addressed in all their parts. The paper can be rejected.
Author Response
Attached please find the response to Reviewer 3

Reviewer 4 Report
The authors have made possible corrections to their manuscript, making it as good as they can. It can now be published in Journal of Composites Science.
Author Response
Attached find the response to Reviewer 4

Round 3
Reviewer 2 Report
Thanks for the clarifications regarding dispersion method.
Author Response
We thank the Reviewer for accepting our amendment
Reviewer 3 Report
Please improve the discussion of mechanical properties (toughness) and consider the possibility to show the stress vs strain curves (even in supporting information).
Author Response
We added comments to the mechanical properties with emphasis on toughness
Round 4
Reviewer 3 Report
the paper can be accepted